# Current Data on Oral Peri-Implant and Periodontal Microbiota and Its Pathological Changes: A Systematic Review

**DOI:** 10.3390/microorganisms10122466

**Published:** 2022-12-14

**Authors:** Virginie Gazil, Octave Nadile Bandiaky, Emmanuelle Renard, Katia Idiri, Xavier Struillou, Assem Soueidan

**Affiliations:** 1Nantes Université, Periodontology Department, CHU (Centre Hospitalier Universitaire) Nantes, UIC Odontology, F-44000 Nantes, France; 2Nantes Université, Oniris, Univ Angers, CHU Nantes, INSERM, Regenerative Medicine and Skeleton, RMeS, UMR 1229, F-44000 Nantes, France

**Keywords:** microbiota, bacteria, dental implant, peri-implantitis, mucositis

## Abstract

The 5- and 10-year implant success rates in dentistry are nearly 90%. Prevalence of peri-implant diseases is 10% for peri-implantitis and 50% for peri-implant mucositis. To better understand these inflammatory pathologies of infectious origin, it is important to know if the composition of the peri-implant microbiota is comparable with the periodontal microbiota in healthy and pathological conditions. New generation sequencing (NGS) is a recent metagenomic method that analyzes the overall microorganisms present in an ecological niche by exploiting their genome. These methods are of two types: 16S rRNA sequencing and the shotgun technique. For several years, they have been used to explore the oral, periodontal, and, more specifically, peri-implant microbiota. The aim of this systematic review is to analyze the recent results of these new explorations by comparing the periodontal and peri-implant microbiota in patients with healthy and diseased sites and to explore the microbiological characteristics of peri-implantitis. A better knowledge of the composition of the peri-implant microbiota would enable us to optimize our therapeutic strategies. An electronic systematic search was performed using the medical databases PubMed/Medline, Cochrane Library, and ScienceDirect, and *Periodontology 2000*. The selected articles were published between January 2015 and March 2021. Inclusion criteria included clinical studies comparing healthy and pathological periodontal and peri-implant microbiota exclusively using 16S rRNA sequencing or shotgun sequencing, with enrolled populations free of systemic pathology, and studies without substantial bias. Eight articles were selected and reviewed. All of them used 16S rRNA sequencing exclusively. The assessment of these articles demonstrates the specific character of the peri-implant microbiota in comparison with the periodontal microbiota in healthy and pathological conditions. Indeed, peri-implant diseases are defined by dysbiotic bacterial communities that vary from one individual to another, including known periodontopathogens such as *Porphyromonas gingivalis* (*P.g.*) and genera less mentioned in the periodontal disease pattern such as *Filifactor alocis*. Examination of peri-implant microbiota with 16S rRNA sequencing reveals differences between the periodontal and peri-implant microbiota under healthy and pathological conditions in terms of diversity and composition. The pattern of dysbiotic drift is preserved in periodontal and peri-implant diseases, but when comparing the different types of pathological sites, the peri-implant microbiota has a specificity in the presence of bacteria proper to peri-implantitis and different relative proportions of the microorganisms present.

## 1. Introduction

Microbiota research began in the 1970s and was enriched by new concepts in the 1990s, such as OTU, bacterial complexes, and quorum sensing. The human mouth harbors one of the most diverse microbiomes in the human body, including viruses, fungi, protozoa, archaea, and bacteria. The oral microbiota contains approximately 700 species, of which individuals have between 100 and 200 [1]. Peri-implant diseases represent a new challenge for the microbiological domain [2]. Ten million dental implants are placed each year worldwide, with a success rate ranging from 82% to over 95% [3]. There are still cases of failure due to lack of osseointegration or as a result of the development of peri-implant diseases such as mucositis (PM) or peri-implantitis (PI) [4]. From a microbiological point of view, these two pathologies seem to have similarities. Indeed, Shi et al. [5] showed in their study that the richness, diversity, and distribution of the microbiome were similar in PM and PI sites, including both common periodontal bacteria and new species. In addition, increased marginal bone loss was significantly associated with submucosal microbial dysbiosis.

Peri-implant mucositis and peri-implantitis are inflammatory diseases of infectious origin, similar to diseases affecting the periodontium. According to the workgroup 4 consensus report from the 2017 World Workshop on the Classification of Periodontal and Peri-implant Diseases and Conditions of the American Academy of Periodontology (AAP) and the European Federation of Periodontology (EFP) [6], mucositis is an “inflammation of the peri-implant soft tissue without bone loss. Typically, mucositis is the precursor to peri-implantitis. It is reversible if intercepted early enough.” Peri-implantitis is defined as an “inflammation of the peri-implant soft tissue with a deterioration of the bone tissue surrounding the tooth” [6].

The implant and the tooth have significant biological, anatomical, and physicochemical differences. These differences could impact the composition of the healthy and dysbiotic microbiota around teeth on one side and implants on the other. Until the mid-2010s, research on the subject highlighted a major microbiological similarity in periodontal and peri-implant diseases [7]. More recently, studies [8,9,10,11] have exclusively started to use new genomic sequencing methods (NGS). It is an innovative approach that allows an unprecedented and comprehensive exploration of the peri-implant microbiota [12].

Previous studies [13,14,15,16,17] have used traditional methods such as culturing, DNA–DNA hybridization, or polymerase chain reaction (PCR). Hybridization and PCR only identified preselected species, usually known periodontal bacteria such as *Prevotella intermedia*, *Aggregatibacter actinomycetemcomitans*, *Porphyromonas gingivalis*, and *Fusobacterium nucleatum*. The drawback of culture methods is that 20% to 60% of the oral microbiota are noncultivable [18] (35% according to another recent article [19]). Among those not-yet-cultivable species are Synergistetes or the TMP bacterial group [20].

The newer genomic sequencing methods have been in existence for approximately 20 years and have overtaken conventional methods. These “new, nontargeted species-oriented methods are currently being used to detect the presence of previously unsuspected or nondescript microorganisms in peri-implant disease scenarios,” the Padial-Molina et al. review points out [18]. NGS, along with 16S ribosomal RNA sequencing and shotgun sequencing, allows for rapid, efficient, and complete detection of microorganisms present in a given environment, without the need to target species [18].

Sequencing of 16S rRNA consists of amplifying 16S rRNA via PCR from the mitochondria of microorganisms prior to a bioinformatics sequencing process [21]. The bacterial genome itself presents great variability within a single species. The 16S rRNA is both universal and very well conserved. Thus, a short region of a gene (generally the V3–V4 region of the 16S gene) is sequenced from all the microorganisms present. This is performed using recent technologies, most often Illumina MiS2016eq. The sequences obtained are identified through similarity with the genomes listed in databases dedicated to the oral microbiota, such as the HOMD (Human Oral Microbiome Database). These small amplified and sequenced sequences are sufficient to identify all the microorganisms present in a given environment, even if their abundance is low [21].

In contrast to 16S rRNA sequencing, shotgun sequencing consists of amplifying the entire genome, but first it is randomly split into shorter fragments. The fragments are cloned, and the clones (called reads) are sequenced. Recognition of the base pairs is also carried out by bioinformatic processing. The different fragments are reassembled to identify the species present. Shotgun sequencing is not yet widely used in the exploration of the peri-implant microbiota, but it is expected to become more widespread. This more exhaustive technique is, however, much more expensive.

Applying these methods to explore the peri-implant microbiota is intended to provide a better understanding of peri-implantitis and whether it is periodontitis-like or different in terms of diversity and composition. The new sequencing methods could also eventually reveal markers of peri-implantitis useful in early diagnosis and implementation of the most appropriate therapies, including the prescription of antibiotics.

The aim of this systematic review is to analyze the recent results of these new explorations by comparing the periodontal microbiota and the peri-implant microbiota in healthy and pathological conditions and to explore the microbiological characteristics of peri-implantitis.

## 2. Materials and Methods

Various studies on the exploration of the peri-implant microbiota with new sequencing methods in comparison with periodontal microbiota were collected and analyzed. PRISMA guidelines (preferred reporting items for systematic review and meta-analysis) were followed and the checklist was completed; it is provided as Appendix A. The PICOS (population, intervention/exposition, comparison, outcomes, and study design) protocol was used to define the selection criteria for eligible articles. Studies that met the following criteria were included:Population: dentate patients with at least 1 implant with a healthy site or with peri-implantitis or mucositis;Exposition: absence or presence of peri-implant disease (peri-implantitis and mucositis);Comparison: pathological and nonpathological implants or pathological implants and pathological or nonpathological teeth;Outcomes: The objectives were to compare the periodontal microbiota and the peri-implant microbiota in healthy and pathological conditions and to explore the microbiological characteristics of peri-implantitis. This was in order to determine the most frequently present and discriminating species of microorganisms by type of site and the key species by type of site and to see if there is a difference in micro-biological diversity;Study design: Clinical case–control studies;Question: Is the peri-implant microbiota different from the microbiota of periodontitis, and what are the characteristics of the peri-implant microbiota?

### 2.1. Information Sources and Search Strategy

The method used was a systematic review of the recent literature (articles published between January 2015 and March 2021) on the subject because the exclusive use of NGS in the exploration of peri-implant microbiota is still new. This article takes over from reviews published since the early 2010s [20,22,23], where the use of NGS already appears in the field of research in peri-implant microbiology but is often mixed with results obtained with traditional methods. The keywords used were determined with the HeTOP glossary of MeSH terms according to the following scheme: (bacteria, NGS) OR (microbiota, periodontal diseases) AND (peri-implantitis) OR (dental implant, heathy site, disease site) OR (mucositis). A systematic search was performed using the PubMed/Medline, Cochrane Library, and ScienceDirect medical databases as well as specialized journals (*Periodontology 2000*).

### 2.2. Article Selection

In each database, the keyword search yielded a first list of results. The title of the articles and/or the abstract were read for a first screening, based on the relevance of the articles to the issue and on the use of NGS. A second screening was performed after reading the articles chosen according to their relevance. Finally, inclusion and exclusion criteria were determined beforehand and applied to establish the list of articles included in this literature review.

### 2.3. Inclusion and Exclusion Criteria

#### 2.3.1. Inclusion Criteria

Articles meeting the following criteria were included: observational case–control studies with two parallel groups or intraoral comparisons, longitudinal cohort studies published between January 2015 and March 2021 with a nonspecified population (without any systemic pathology); clinical trials involving in vivo implants placed in humans; 16S ribosomal RNA or shotgun sequencing; comparison of sites with periodontitis/peri-implantitis or mucositis/gingivitis and healthy peri-implant areas or presenting with mucositis/peri-implantitis; high or satisfactory level of evidence-based studies with internal and external validity criteria; and assessment of risk of bias.

#### 2.3.2. Exclusion Criteria

Studies concerning culturing with species counting and DNA–DNA hybridization; targeting a limited number of known bacterial species; comparing microbiota in smokers versus nonsmokers; implants placed less than 6 months ago; too small a population; in vitro studies or animal studies or experimental titanium devices simulating an implant; meta-analyses; literature reviews; editorials; expert opinions; and case series were excluded.

### 2.4. Data Collection

The following data on the characteristics of each study were recorded by the same author (V.G.) and controlled by another reviewer member (X.S.): (1) type of study: cohort, case–control; (2) population; (3) study design (composition of the study groups); (4) method of exploration of the microbiota; (5) *p* values of the statistical tests; (6) results (diversity of the microbiota by type of site, association of the microorganisms with clinical data (unaffected implant, pathological implant, healthy periodontitis)).

### 2.5. Risk of Bias of Each Study and Data Synthesis

The tool used for risk of bias assessment was the Joanna Briggs Institute (JBI) critical appraisal checklist for case–control studies, last updated in 2017 [24]. The JBI’s critical appraisal tools are intended to assess methodological quality and the extent of bias in the design, conduct, and analysis of data. The JBI has checklists for critical appraisal of prevalence, cohort, and case-controlled studies, as well as RCTs. Therefore, the choice of the JBI critical appraisal checklist is justified in our review as it relates to case-controlled studies. A descriptive and systematic review of articles was performed. All studies included except one were case–controls [11]. This one was presented by the authors as using a previously formed cohort. However, the study design used is a case-controlled comparison.

## 3. Results

### 3.1. Study Characteristics and Summary of Results

The bibliographic search identified 13282 articles, of which 12209 were excluded after reading the titles/abstracts. Full-text reading of 73 eligible articles excluded 65 and included 8 [9,10,11,25,26,27,28,29]. The characteristics of the included studies are presented in Table 1. The flowchart of the review is presented in Figure 1. Seven studies [9,10,25,26,27,28,29] were case–control studies, and one [11] was a longitudinal study that randomly selected patients from a cohort formed during the 15 previous years for case–control comparisons.

The population included varied from 10 to 67 subjects depending on the study, and age varied from 21 to 86 years old. Two of the articles did not mention the age of the population [9,29], and one article [25] only provided information about the mean age of the two groups. All microbiota measurements, including probing, bleeding on probing (BOP), attach loss, and plaque sample collection were performed on implants and teeth, with at least one group having peri-implantitis. Four studies [26,27,28,29] were intraindividual comparisons: the peri-implantitis groups and the healthy control groups were from different sites in the same patients. Three of the other studies [9,10,25] compared different populations in several distinct groups. The study by Sousa et al. [11] mixed these designs: five groups present five different conditions with different populations in each, whereas the last group was a control group including some of the patients of the other groups [11]. Two studies [11,25] included patients who smoked but we did not distinguish them into different groups because this review did not address the effect of smoking on the peri-implant microbiota. The eight included articles exclusively used 16S rRNA sequencing for exploration of the microbiota. Data analysis showed that there were differences between the microbiota of periodontitis and the microbiota of peri-implantitis. The implant constitutes a distinct environment; peri-implantitis is characterized by a very variable microbiota among individuals but is rich in gram-negative periodontitis pathogens (*Porphyromas gingivalis, Treponema denticola, Fusobacterium nucleatum*) as well as by diseased-implant-specific bacteria (*Filifactor alocis*) that do not appear in the periodontitis dysbiotic pattern or in the healthy implant microbiota. Metagenomics reveals a richness in species of peri-implantitis microbiota and a great variability between individuals that traditional methods would not have detected. Data on the characteristics of the implants (brand, size, diameter, type of implant), the location of the implants (maxillary/mandibular, anterior/posterior), the date of loading, and the date of bacterial sampling on the implant site between the placement and the patient’s visit were rarely reported by the authors.

### 3.2. Risk of Bias Analysis with the Joanna Briggs Institute Case–Control Study Checklist (2017) [24]

The analysis of risk of bias (Figure 2) shows that the risk of selection bias was low overall, but a few studies did not mention the inclusion and exclusion criteria for the population included in the groups, making it impossible to judge the homogeneity of the groups. A performance bias was present in most of the studies [9,25,27,28,29] because the authors did not provide information on the experimenters or the calibration of the measurements. Only three studies had a clear and explained protocol to counteract this bias. The comparability of data between the case and control groups was very good, and no bias was detected. The results were clear and precise, although two studies [10,28] had less-developed analyses [10,28]. Confounding bias was detected for studies including smokers and nonsmokers [11,25], as no statistical correction was reported.

## 4. Discussion and Conclusions

Ten million dental implants are placed each year, with a success rate over 90% [3]. There are still cases of failure due to a lack of osseointegration or the development of peri-implant disease. The aim of this systematic review is to compare the periodontal and peri-implant microbiota in healthy and pathological conditions and to explore the microbiological characteristics of peri-implantitis via 16S rRNA sequencing and the shotgun technique.

In this work, the authors of the eight included studies [9,10,11,25,26,27,28,29] agreed that peri-implantitis has a different microbiota than periodontitis and is a distinct pathological entity. Six included studies addressed the diversity of the microbiota according to ecological niches, but no consensus emerged: two studies [10,29] concluded that there was no difference in terms of diversity and two studies [11,26] concluded that healthy sites had a richer microbiota than that associated with peri-implantitis. Two other studies [9,25] tended to find more microbiological diversity associated with peri-implantitis. These results are consistent with the findings of the systematic review by other authors [12,30].

In the articles reviewed in this study, the bacterial species most frequently associated with peri-implantitis are known pathogens (*P. gingivalis*, *F. nucleatum*, *T. forsythia*), but they were not identified in all studies, and there was no consensus. On the other hand, there was a great variability of results between studies, with many species cited in one or two articles. De Melo et al.’s review [12] produced similar results. Some species are known to be periodontopathogenic, but our analysis indicates that it has not been possible to determine a systematic bacterial marker for peri-implantitis. Rather, it is the result of a variable dysbiotic community among individuals.

The composition of the periodontal and peri-implant microbiota was analyzed to highlight the particularities of each type of site. Many bacterial species are common to all sites (periodontal, peri-implant, healthy, or pathological) but in different proportions. We will refer to those species that are more abundantly detected in a specific site within each study as discriminating species. The discriminating species associated with peri-implantitis cited in at least two studies are species of the red complex: *P. gingivalis* [10,25,26,28,29], *Treponema denticola* [25,29], and *Tanerella forsythia* [25,28], and species of the orange complex: *P. intermedia* [26,29], Bacteroides [25,26,28], and *Filifactor* sp. [11,25,29]. These articles are similar to previous works, as reported in an article by Charalampakis and Belibasakis published in 2015 [7]. The pathogens *F. nucleatum*, *P. intermedia*, and *P. gingivalis* are also cited in the studies included here, but some authors conclude that these microorganisms coexist in both healthy and pathologic implant sites [9], with a higher relative abundance in affected sites.

Healthy implant sites are specifically associated with *Rothia* [9,10,25], *Neiserria* [9,25,29], and *Corynebacterium* [9,11]. Methanogens are a part of the commensal periodontal and peri-implant flora and are present in both healthy and diseased sites, without being site-specific [27]. According to the results obtained by Belkacemi et al. [27], they are not part of the dysbiotic pattern and are not correlated with peri-implantitis. The oral microbiota is rich and diverse, but the microbial drift associated with periodontal and peri-implant pathologies is almost exclusively bacterial.

A number of the studies presented here are intraoral comparisons. This is a very interesting procedure that both highlights interindividual variability and circumvents it to obtain a site-specific comparison within the same individual. The results obtained from two studies challenge dental reservoir and germ translocation theory [10,11]. In the Sousa et al. study [11], differences in microbiota composition were obtained intraorally by comparing sites with peri-implantitis to implants without lesions or to natural teeth without periodontal lesions. In contrast, the Yu et al. study [29], which compared four different types of sites per patient, concluded that interindividual variability was important. In this study, very few significant differences were found according to the type of site (no significant difference when comparing implant and tooth, some differences when comparing healthy or pathological sites), but it seems that each individual has a different microbiota.

Certain species are more frequently associated with peri-implantitis, and the dysbiotic pattern of periodontal disease is retained in peri-implantitis. However, rather than targeting a few species, it is more appropriate to think of peri-implant disease in terms of bacterial communities that promote dysbiosis, while other bacterial communities promote health. The dysbiotic structure remains unchanged, but the microbiota associated with dysbiosis and symbiosis offer variable characteristics. Thus, species considered to be periodontopathogenic are also found in healthy periodontal and peri-implant sites [9,25,26,29].

Peri-implant mucositis is considered a precursor stage of peri-implantitis. Its microbiota consists of both bacterial species that are more present in healthy sites and species that are more prevalent in sites with peri-implantitis [9].

The results obtained by the new exploratory metagenomic methods make it possible to obtain a more complete and exhaustive vision of the analyzed samples. Indeed, with traditional methods, researchers have concluded that the microbiological profile of peri-implant diseases is equivalent to that of periodontal diseases [7]. The peri-implant areas have a distinct ecosystem from the periodontium. Their microbiota show notable differences. The etiology of peri-implant diseases is polybacterial. It results from the complex relationships between the host (anatomical niche, immune response) and the bacterial communities. The role of the known periodontal pathogens must be put into perspective. The theory of a simple translocation of periodontal microorganisms to the peri-implant areas is not sufficient to understand the implant microbiota. The peri-implant microbiota presents many unknowns. Although nonabundant species are much more easily detected, their role has yet to be elucidated. For an equivalent amount of biofilm, is bacterial diversity a sign of symbiosis or dysbiosis? The small number of studies that could be included in this work does not allow us to answer this question for the moment. The importance of interindividual variability questions the relevance of this often-discussed question.

The peri-implant microbiota is unique and complex: it consists of many species, mostly bacterial. Many species present in low abundance have been detected and identified by NGS. These species, previously not associated with the periodontium or the peri-implant environment, could play a role in the onset or progression of diseases.

The eight included studies presented heterogeneous results. Some studies included a small number of patients, which may contribute to these discrepancies, limiting the ability to generalize the data obtained. A systematic review by De Melo et al. [12] on the same topic was published in 2020. This systematic review included seven studies and made the same observation of the heterogeneity of the available studies in terms of the characteristics of the populations included, the brand and type of implant, the date the implant was put into use, or the design of the studies. Of the seven articles included by De Melo et al. [12], four were articles [9,11,25,29] also used in this work. As mentioned above, there are three other recent systematic reviews of the literature also comparing periodontal and peri-implant microbiota, but their scope is broader and includes studies using traditional microbiological analysis methods [20,22,23]. Thus, the possibility of comparison with the above three studies is limited.

NGS revealed an unanticipated diversity and richness of the periodontal and peri-implant microbiota. Yu et al. [29] detected 5726 OTUs, and Zheng et al. [9] detected 15,766 OTUs across the different sites. Some bacterial species detected thanks to metagenomics were never previously associated with the periodontal and peri-implant microbiota, and methanogens (*Methanobrevibacter oralis*, *Methanobrevibacter massiliense*) are part of the commensal and normal flora of the oral cavity [27]. Thus, “an unclassified Mycoplasma [is] positively correlated to BOP and periodontal probing depth (PPD)” on implants according to the Yu et al. study [29]. The peri-implant microbiota was found to be less specifically related to certain distinctive bacterial species (the red complex by Socranky) than earlier studies had suggested [9], but it was also more diverse in species. Many noncultivable species present in sparse abundance have been identified, and species never previously detected in these sites have been discovered, such as *Helicobacter pylori* [28]. Recent studies using metagenomic methods highlight the interindividual variability of the peri-implant microbiota and the involvement of environmental factors. As De Melo et al. explain, “16s rRNA gene sequencing can provide a large proportion of the microorganisms detected in low abundance and thus contribute to the high diversity and interindividual variability of the oral microbiota” [12].

Future work using 16S rRNA or shotgun sequencing techniques with larger and homogenous (sex, age, medical history) population inclusions is recommended, being more precise about the specificities of the implants, with a cross-sectional case–control design and a calibration of data collection. This will provide insight into the specific nature of the peri-implant microbiota.

Current research focuses on the physicochemical properties of implants, including surface coatings and the use of antimicrobial peptides. Some authors, such as Zhang et al. [31], have shown in their studies that tantalum-modified titanium (Ti) implants exhibited excellent antimicrobial activity against *Fusobacterium nucleatum* and *Porphyromonas gingivalis*. However, peri-implant pathology progresses more rapidly than periodontitis for anatomical and histological reasons. Therefore, it is essential to identify the specificity of the microbiota associated with peri-implant disease but also with the healthy peri-implant area and, from a clinical point of view, to propose the most suitable implants with the aim of inhibiting the establishment and progression of peri-implant disease.

## Figures and Tables

**Figure 1 microorganisms-10-02466-f001:**
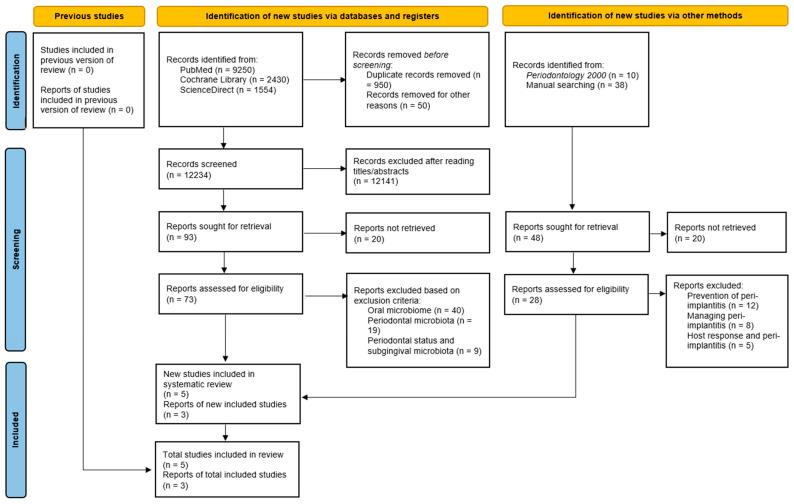
Flow diagram of the current systematic review.

**Figure 2 microorganisms-10-02466-f002:**
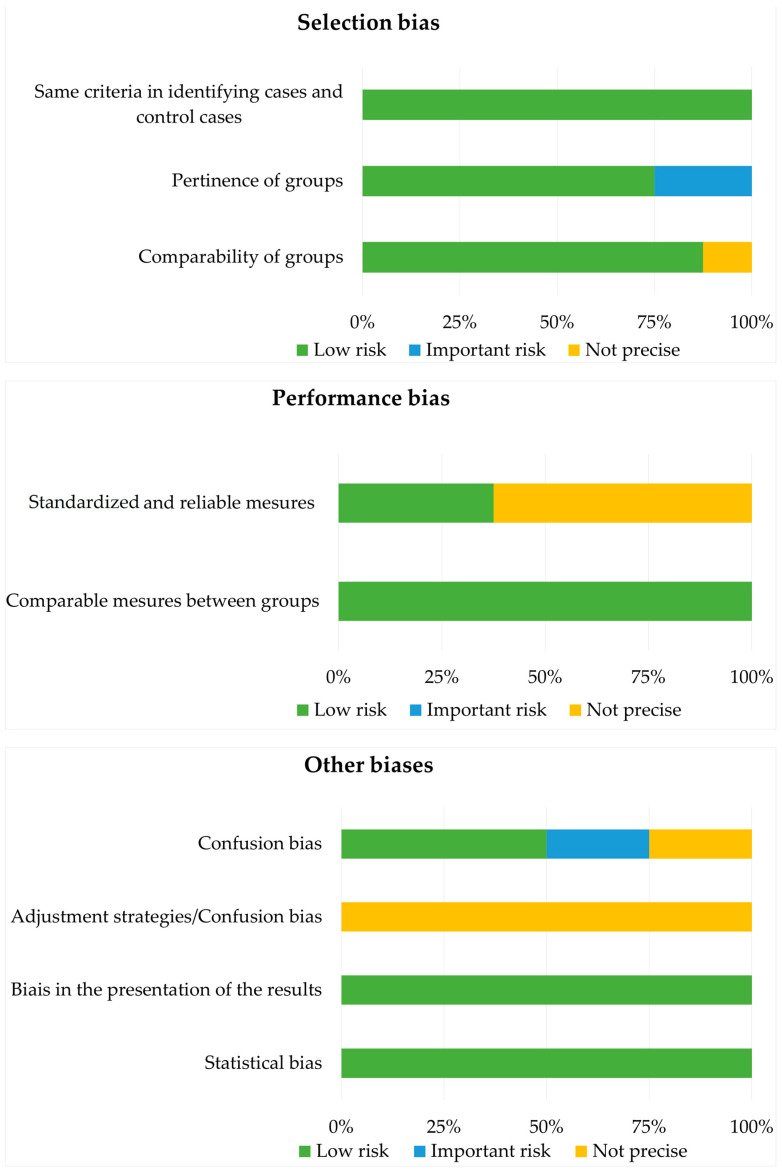
Risk of bias analysis with the Joanna Briggs Institute case–control study checklist.

**Table 1 microorganisms-10-02466-t001:** Characteristics of the included studies.

Author, Year	Study Design	Participants	Study Group	Exploration Method	Results
Zheng, H. et al., 2015 [9]	case–control	24	− peri-implantitis group − control group (nonaffected implant) − mucositis group	16S rRNA sequencing	Peri-implantitis is associated with more diversity of the microbiota. Healthy and pathological sites have distinct bacterial communities. Peri-implantitis is associated with *Firmicutes*, *Fusobacteria*, *Proteobacteria*, and *Actinobacteria*.
Jakobi, M. et al., 2015 [10]	case–control	18	− group with peri-implantitis − control group with nonaffected implant − group with periodontitis − group with unaffected tooth	16S rRNA sequencing	Tendency to more diversity in affected sites without significant difference; sites with peri-implantitis associated with *Neisseria* and *Kingella*; sites with periodontitis associated with *Rothia*, *Tannerella*, and *Parabacteroides*; pathological sites (tooth and implant) associated with *Enterococcus*, *Streptococci*, *Porphyromonas gingivalis*, *F. nucleatum*, *Fretibacterium*, *P. intermedia*, and *Bacillus*
Sousa, V. et al., 2017 [11]	case–control	24	− 3 groups with implant: healthy, mucositis and peri-implantitis − 3 periodontal control groups: healthy and pathological	16S rRNA sequencing	More microbial diversity at dental sites than at implant sites; some bacteria are implant-specific: Propionibacteria, *Paludibacter, Staphylococci, Filifactor,* and *Mogibacterium*; dominant genus of peri-implantitis: *Firmicutes*, dominant genus of nonpathological implant: *Streptococci*
Sanz-Martin, I. et al., 2017 [25]	case–control	67	− peri-implantitis group − control group with nonaffected implant	16S rRNA sequencing	Different pathological and nonpathological peri-implant microbiota; peri-implantitis is associated with more microbiota diversity; peri-implantitis is associated with the red complex and new pathogens: *Filifactor alocis, Fretibacterium fastidiosum*, and *Treponema maltophilum*; healthy sites are associated with *Streptococci, Rothia*, and *Haemophilus*
Apatzidou, D. et al., 2017 [26]	case–control	10	− peri-implantitis group − control group with nonaffected implant	16S rRNA sequencing	*Methanobrevibacter oralis* is present in more than 50% of the samples; no significant difference between the 2 groups; no association between *Methanobrevibacter* and peri-implantitis.
Belkacemi, S. et al., 2018 [27]	case–control	28	− peri-implantitis group − control group with nonaffected implant	16S rRNA sequencing	Different pathological and nonpathological peri-implant microbiota; bacterial drift associated with peri-implantitis with increased presence of the red complex; peri-implantitis is associated with Bacteroides (*F. nucleatum*), *Porphyromonas gingivalis*, and *T. forsythia*
Al-Ahmad, A. et al., 2018 [28]	case–control Intraoral comparison	10	− peri-implantitis group − control group with nonaffected implant	16S rRNA sequencing	Different pathological and nonpathological peri-implant microbiota; bacterial drift associated with peri-implantitis with increased presence of red complex; peri-implantitis is associated with Bacteroides (*F. nucleatum*), *P. gingivalis* and *T. forsythia.*
Yu, X. et al., 2019 [29]	case–control Intraoral comparison	18	− peri-implantitis group − control group with nonaffected implant − periodontitis group − group with nonaffected tooth	16S rRNA sequencing	Interindividual variability is more important than the variability related to pathological presence; peri-implantitis is characterized by *P. gingivalis, T. forsythia, A. actinomycetemcomitans, Treponema* spp.; and with rare species present in the mouth: *Staphylocoques, Peptostreptococci, Enterobacteries, Helicobacter* spp.

## Data Availability

Not applicable.

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
