# Peer review of "Current Data on Oral Peri-Implant and Periodontal Microbiota and Its Pathological Changes: A Systematic Review"

_microorganisms, 2022, doi:10.3390/microorganisms10122466_

Round 1

Reviewer 1 Report

I must congratulate the authors for the manuscript. A well-conducted systematic review, with a broad spectrum of searches. The Prisma flow-shart and the analysis of bias are well done. The discussion and conclusions are integrated in the parameters evaluated.

Studying the peri-implant microbiota is crucial to implement treatment guidelines, but still not certified in sound research.

Author Response

Dear Reviewers,

Thank you for the reviews of our manuscript Microorganisms-2069502- and for allowing us to re-submit another version of this work describing “Current data on oral peri-implant and periodontal microbiota and its pathological changes: A systematic review.”  We very much appreciate the constructive comments of the reviewers and have addressed all their points below in a variety of ways.  Thanks to these comments, we feel that the manuscript has indeed been improved – thank you.        

Responses to the reviewers’ verbatim comments are listed below and in yellow in the revised manuscript in enclosed file.

Pr A. Soueidan

Reviewer 2 Report

This literature review entitled "Current data on oral peri-implant and periodontal microbiota and its pathological changes: a literature review" aims to analyze the recent results of these new explo-rations by comparing the periodontal microbiota and the peri-implant microbiota in healthy and pathological conditions and to explore the microbiological characteristics of peri-implantitis.

Advances in the field of microbiological analysis techniques have enabled a true paradigm shift in the understanding of the periodontal microbiota physiological or associated with periodontal disease. These techniques bring real advances also concerning the peri-implant flora. The manuscript is well written and presents pertinent scientific information.

The authors concluded that the pattern of dysbiotic drift is preserved in periodontal and peri-implant diseases, but when comparing the different types of pathological sites, the peri-implant microbiota has a specificity with the presence of bacteria proper to peri-implantitis and different relative proportions of the microorganisms present

Per se, this article is of interest. However, I will make some suggestions to improve the clarity and understanding of the manuscript.

Abstract:

Regarding the inclusion criteria, it should be specified that the in vivo studies included only concern implants in humans and not studies in animals.

Material and methods

The PICOS method described is a little complex, we do not really find the objectives, however clearly stated in the abstract. For example "Outcomes: The objectives are to determine the most frequently present and discrim-inating species of microorganisms by type of site and the key species (which induce pa-thology even in small quantities) by type of site and to see if there is a difference in micro-biological diversity."

In information sources and search strategy, It is written that "A systematic search was performed"but this is the only place where the systematic mention appears while the methodology is good. Could you explain it?

For Risk of Bias assessment, the choice of the Joanna Briggs Institute (JBI) Critical Appraisal Checklist for case‒control study should be justified and referenced. Why this one rather than the Newcastle Ottawa scale for example ?

Discussion

This sentence "It should be noted that no studies including viruses are presented here, as there is little published work on this topic [30]". If we focus on studies using the 16s RNA sequencing technique, it seems quite logical that no viruses appear since this technique sequences the ribosomal RNA of bacteria.

Author Response

Dear reviewers,

Thank you for the reviews of our manuscript Microorganisms-2069502- and for allowing us to re-submit another version of this work describing “Current data on oral peri-implant and periodontal microbiota and its pathological changes: A systematic review.”  We very much appreciate the constructive comments of the reviewers and have addressed all their points below in a variety of ways.  Thanks to these comments, we feel that the manuscript has indeed been improved – thank you.

Responses  comments are listed below and in yellow in the revised manuscript in enclosed file.

Regards

Pr A. Soueidan
